# *MALINC1* an Immune-Related Long Non-Coding RNA Associated with Early-Stage Breast Cancer Progression

**DOI:** 10.3390/cancers14122819

**Published:** 2022-06-07

**Authors:** María Laura Fabre, Romina Canzoneri, Agustina Gurruchaga, Jaeho Lee, Pradeep Tatineni, Hyunsuk Kil, Ezequiel Lacunza, C. Marcelo Aldaz, Martín Carlos Abba

**Affiliations:** 1Centro de Investigaciones Inmunológicas Básicas y Aplicadas (CINIBA), Facultad de Ciencias Médicas, Universidad Nacional de La Plata, La Plata 1900, Argentina; marialaura.fabre@biol.unlp.edu.ar (M.L.F.); romac@med.unlp.edu.ar (R.C.); agustinagurruchaga@hotmail.com (A.G.); elacunza@med.unlp.edu.ar (E.L.); 2Department of Epigenetics and Molecular Carcinogenesis, The University of Texas MD Anderson Cancer Center, Houston, TX 77054, USA; jaehole@yahoo.com (J.L.); tatinenip@icloud.com (P.T.); hkil@mdanderson.org (H.K.)

**Keywords:** *MALINC1*, lncRNA, breast cancer, DCIS

## Abstract

**Simple Summary:**

Here we characterize the phenotypic and molecular effects of *MALINC1*, a long non-coding RNA (lncRNA) that we found significantly upregulated in premalignant ductal carcinoma in-situ lesions. We provide evidence that *MALINC1* behaves as an oncogenic and immune-related lncRNA involved with early-stage breast cancer progression, showing prognostic and predictive value to immunotherapy in invasive breast carcinomas.

**Abstract:**

Long non-coding RNAs are increasingly being recognized as cancer biomarkers in various malignancies, acting as either tumor suppressors or oncogenes. The long non-coding *MALINC1* intergenic RNA was identified as significantly upregulated in breast ductal carcinoma in situ. The aim of this study was to characterize *MALINC1* expression, localization, and phenotypic and molecular effects in non-invasive and invasive breast cancer cells. We determined that *MALINC1* is an estrogen–estrogen receptor-modulated lncRNA enriched in the cytoplasmic fraction of luminal A/B breast cancer cells that is associated with worse overall survival in patients with primary invasive breast carcinomas. Transcriptomic studies in normal and DCIS cells identified the main signaling pathways modulated by *MALINC1*, which mainly involve bioprocesses related to innate and adaptive immune responses, extracellular matrix remodeling, cell adhesion, and activation of AP-1 signaling pathway. We determined that *MALINC1* induces premalignant phenotypic changes by increasing cell migration in normal breast cells. Moreover, high *MALINC1* expression in invasive carcinomas was associated with a pro-tumorigenic immune environment and a favorable predicted response to immunotherapy both in luminal and basal-like subtypes compared with low-*MALINC1*-expression tumors. We conclude that *MALINC1* behaves as an oncogenic and immune-related lncRNA involved with early-stage breast cancer progression.

## 1. Introduction

Breast cancer has become the most common cancer worldwide, contributing to 12% of the total number of new cases diagnosed in 2020 [1]. Invasive ductal breast carcinoma (IDC) is the most frequent malignancy of the breast, accounting for ~80% of all invasive breast tumors, according to the American Cancer Society’s Cancer Statistics Center [2]. Ductal carcinoma in situ (DCIS), also known as intraductal carcinoma, is by definition a cancerous precursor lesion to IDC with no regional or lymph node involvement. Retrospective epidemiological studies have concluded that women with biopsy-proven DCIS have over a 10-fold higher risk for developing invasive breast cancer than women without history of these lesions [3,4]. However, the reasons for why only some DCIS lesions progress to the invasive stage remain unclear, and almost all women with biopsy diagnosed DCIS undergo some form of treatment; thus, accurate prediction of the likelihood of progression is needed to avoid over-treatment.

Non-coding RNAs (ncRNA) are of crucial relevance in many important biological processes [5,6,7]. In particular, the class of transcripts known as long non-coding RNAs (ncRNAs >200 nt long) have been recognized to be tissue and cell type-specific, playing key roles in regulating chromatin dynamics, gene expression, growth, and differentiation [8,9]. Aberrant expression of long non-coding RNAs (lncRNAs) in human cancers has been increasingly reported in recent studies, suggesting a promising role as diagnostic and prognostic biomarkers [10,11]. Furthermore, deregulation of specific lncRNAs has been demonstrated to be closely related to the development and progression of individual breast cancer subtypes. They were shown to act as promoters or inhibitors of breast cancer progression, modulating the proliferation, apoptosis, epithelial mesenchymal transition, metastatic dissemination, and drug resistance of cancer cells. Examples of this include oncogenic lncRNAs such as *MALAT1*, *NEAT1*, *H19*, and *HOTAIR* or as tumor suppressive counterparts such as *MEG3*, *XIST*, and *PTENP1* [11].

In a previous study, we performed the first comprehensive molecular profiling of “pure” DCIS lesions, identifying, among other genomic abnormalities, hundreds of lncRNAs with deregulated expression, many of which might be associated with breast cancer progression [12]. In more recent studies we characterized novel lncRNAs such as *LINC00885*, and previously known lncRNAS like *HOTAIR*, for their role as inducers of pro-oncogenic changes in normal and premalignant breast cells [13,14]. 

In the original studies we identified *LINC01024* as one of the significantly upregulated lncRNAs in breast DCIS lesions [12]. In contemporaneous studies, *LINC01024* was associated with cell-cycle progression of osteosarcoma cells and renamed *MALINC1* (*Mitosis-Associated Long Intergenic Non-Coding RNA 1*). In the same study it was also reported that high *MALINC1* expression correlated with poor overall survival in breast and lung cancer patients, and this might be related to the inhibition of anti-mitotic drug paclitaxel-induced apoptotic cell death [15]. More recently, *MALINC1* was defined as an immune-related lncRNAs significantly associated with clinical outcomes in patients with melanoma [16].

Here, we characterized for the first time the expression, molecular and phenotypic effects of *MALINC1* in non-invasive and invasive breast cancer models. Overall, we obtained strong evidence to propose this long non-coding RNA as a novel early-stage breast cancer-associated gene.

## 2. Materials and Methods

### 2.1. Cell Lines, Cell Culture

The MCF10A cell line was obtained from the American Type Culture Collection (#CRL-10318; ATCC, VA, USA) and validated by DNA fingerprinting. MCF10A cells were cultured in Dulbecco’s Modified Eagle Medium F-12 (DMEM/F-12, Sigma-Aldrich, St. Louis, MO, USA) supplemented with 5% horse serum, 20 ng/mL epidermal growth factor (Sigma-Aldrich), 100 µg/mL hydrocortisone (Sigma-Aldrich), 10 µg/mL insulin (Sigma-Aldrich), 100 ng/mL cholera toxin (Sigma-Aldrich), and 100 U/mL penicillin–100 μg/mL streptomycin (Sigma-Aldrich). MCF10 DCIS.COM (hereafter DCIS.COM) cells were a kind gift from Dr. Daniel Medina and were maintained in DMEM/F-12 supplemented with 5% horse serum. The MCF7 and T47D cell lines were cultured in DMEM (Sigma-Aldrich, St. Louis, MO, USA) supplemented with 10% fetal bovine serum (FBS; Natocor, Córdoba, Argentina) and 100 U/mL penicillin–100 μg/mL streptomycin (Sigma-Aldrich, St. Louis, MO, USA). Cell lines were maintained at 37 °C with 5% CO_2_.

### 2.2. LncRNA Subcellular Localization

To determine the subcellular localization of endogenous *MALINC1* transcripts in breast cancer lines we used the PARIS Kit (Thermo Fisher Scientific, Waltham, MA, USA) to isolate nuclear and cytoplasmic RNA fractions from cultured cells. Briefly, T47D and MCF7 cells were cultured in 10 mm plates as described above. Cells were trypsinized, washed, and resuspended in the required amount of lysis buffer. The nuclear and cytoplasmic fractions were separated by centrifugation at 400× *g* for 1 min at 4 °C. RNA was extracted from the two fractions according to the kit instructions. The expression of *MALAT1* and *MTRNR1* were determined by RT-qPCR in each fraction as nuclear and cytoplasmic markers, respectively.

### 2.3. Stable MALINC1-Expressing Cells

The full-length sequence of *MALINC1* (5090 bp spanning three exons, NCBI Entrez Gene: 100505636, Ensembl Gene ID: ENSG00000245146, Ensembl Transcript ID: ENST000004992037) was synthesized (Genscript, Piscataway, NJ, USA), sequence verified, and subsequently cloned into the pLOC lentiviral expression vector. Virus particles were produced using packaging line Lenti-X 293T (Takara Bio, Mountain View, CA, USA). Normal breast epithelial cell lines MCF10A and DCIS cell line DCIS.COM were stably transduced and selected with 10 µg/mL blasticidin. *MALINC1* overexpression was confirmed in all cell lines by RT-qPCR.

### 2.4. Cell Proliferation, Motility, and Migration Assays

MCF10A stably transduced to overexpress *MALINC1* or an empty vector control were plated (1000 cells per well) on 96-well plates in triplicate and cell proliferation was determined by means of the colorimetric MTT assay kit (Cell Proliferation Kit, Roche) and measuring optical density (OD). To assess cell motility, we conducted a standard wound-healing assay. Briefly, 1 × 10^6^ cells were seeded in each well. After cells adhered, the FBS concentration in the medium was reduced to 2% to decrease cell proliferation. Two scratch wounds were made in confluent cell cultures. Images of the same fields were collected at 0, 24, and 48 h. To quantify the cell migration rate, the width of the wound was determined at 10 separate sites for each time point. The assay was performed in triplicate, and the mean and standard deviation were calculated for each determination. Transwell migration assays were performed using standard Boyden chambers containing 12 μm pore divider membranes, and 5% FBS was used in the lower chamber as a chemoattractant. Statistical significance was determined using the Mann–Whitney–Wilcoxon test. 

### 2.5. Estradiol Induction Assay

MCF7 cells were cultured in RPMI medium without phenol and without fetal bovine serum (FBS; Natocor, Córdoba, Argentina) and 100 U/mL penicillin–100 μg/mL streptomycin (Sigma-Aldrich, St. Louis, MO, USA) for 3 days. After that estradiol was added at a final concentration of 100 mM. Cells were harvested after 1, 3, and 6 h post estradiol induction. MCF7 cells without estradiol treatment were used as a negative control. *XBP1* was used as a reference gene for response to estradiol induction and the expression of *MALINC1* was measured by RTq-PCR.

### 2.6. RNA Isolation and Real-Time Quantitative PCR (RT-qPCR)

Total RNA was isolated from cell lines using TRI Reagent solution (Thermo Fisher Scientific, Waltham, MA, USA) according to the manufacturer’s instructions. RNA was reverse transcribed into cDNA using the SuperScript reverse transcriptase kit (Thermo Fisher Scientific, Waltham, MA, USA). Primers were designed for *MALINC1*, *XBP1*, *MALAT1*, *MTRNR1*, *FOS*, *JUN*, *GAPDH*, and *RNA18S* (Appendix A). PCR conditions were as follows: An initial denaturation step of 95 °C for 3 min and 40 cycles of 95 °C 40 s, 55–60 °C 30 s, and 72 °C 30 s was carried out. Data were captured and analyzed using the AriaMx Real Time PCR software (Agilent Technologies, Santa Clara, CA, USA). Real-time PCR assays were performed using PerfeCTa SYBR Green SuperMix (Quanta BioSciences Inc., Beverly Hills, CA, USA). Experiments were done in triplicate and normalized to *GAPDH* or *RNA18S* expression using the comparative threshold cycle (2^−ΔΔCt^) method. Gene expression levels were calculated as 2^−ΔΔCt^ values using the housekeeping genes *RNA18S* or *GAPDH* as a reference. The relative expression units among groups were compared using the Mann–Whitney U test in R software. *p* < 0.05 was considered to indicate a statistically significant difference. Correlation analyses of *MALINC1*, *FOS*, and *JUN* transcripts among breast tissue samples were performed based on their discretized relative expression values (0: undetectable, 1: low expression, 2: high expression) by Kendall’s test.

### 2.7. RNA-seq Data Analysis

MCF10A and DCIS.COM stably transduced cells were used for RNA isolation from subconfluent plates using the RNeasy kit (Qiagen, CA, USA). RNA concentration and integrity were measured on an Agilent 2100 Bioanalyzer (Agilent Technologies). Only RNA samples with RNA integrity values (RIN) over 8.0 were considered for subsequent analysis. RNA-seq library construction was performed using the ScriptSeq v2 RNA-seq Library Preparation Kit (Epicentre) according to the manufacturer’s protocol. We performed 76 nt paired-end sequencing using an Illumina HiSeq2000 platform and ~20 million reads per sample were obtained. The short-sequenced reads were mapped to the human reference genome (hg19) by the splice junction aligner Rsubread package. We employed several R/Bioconductor packages to accurately calculate the gene expression abundance at the whole-genome level using the aligned records (BAM files) and to identify differentially expressed genes between cells stably transduced with *MALINC1* and empty vector. Briefly, the number of reads mapped to each gene based on the UCSC.hg19.KnownGene database were counted, reported, and annotated using the featureCounts and org.Hs.eg.db packages. Data are available at GEO under accession number GSE194150. To identify differentially expressed genes (false discovery rate (FDR) < 0.01; fold change (FC) > ±2) between the empty vector and *MALINC1*-overexpressing counterparts, we utilized the edgeR Bioconductor package based on the normalized log2 based count per million values. For functional enrichment analyses, we used the R/Bioconductor packages clusterProfiler and enrichplot. 

Pathway-based analysis was performed using the PARADIGM software at the Five3 Nantomics server on the basis of the normalized gene expression profiles of the MCF10A and DCIS.COM count matrix (log2CPM) [17]. PARADIGM produces a data matrix of integrated pathway activities (IPAs). This data matrix was used in place of the mRNA expression profiles to identify the topmost variable IPAs among MCF10A and DCIS.COM *MALINC1*-expressing cells using the rank product test (p-adj. < 0.01). Heatmap visualization of differentially expressed transcripts and IPAs was carried out with the MultiExperiment Viewer software (MeV v4.9).

### 2.8. In Silico Analysis of MALINC1 Expression in Normal and Breast Cancer Samples

Pre-processed *MALINC1* expression profiles among two early-stage breast cancer datasets—GSE69994 and GSE169393—were obtained from GEO and analyzed using R software. In addition, pre-processed *MALINC1* RNA-seq expression levels among primary breast carcinomas with intrinsic subtype data were obtained from the TCGA Breast Cancer (BRCA) dataset through the UCSC Xena browser (http://xena.ucsc.edu/) (accessed on 12 October 2021). Primary breast carcinomas (*n* = 1097) were divided into low (*n* = 153) or high (*n* = 307) *MALINC1* expression levels according to the StepMiner one-step algorithm [18]. These two groups were then compared to calculate the percentage of cases with high or low *MALINC1* expression among intrinsic subtypes. For overall survival analysis, breast cancer patients with luminal A/B primary carcinomas (*n* = 310) were grouped into high or low *MALINC1* cases. Statistical analysis was performed using the R packages survival and survminer. Immune cell fractions were estimated using the quanTIseq deconvolution algorithm implemented in the immunedeconv R package based on RNA-seq profiles of high or low *MALINC1* primary invasive breast carcinomas [19]. The Tumor Immune Dysfunction and Exclusion score (TIDE) was used to predict the immune checkpoint blockade response of patients with high or low *MALINC1* breast carcinomas [20].

## 3. Results and Discussion

### 3.1. MALINC1 Is Overexpressed in Luminal Breast Cancer Subtypes and Associated with Poor Prognosis 

In silico analysis of *MALINC1* expression in early-stage breast cancer datasets obtained from Gene Expression Omnibus (GEO) showed significant upregulation of this transcript in DCIS when compared to normal samples (*p* = 0.0004 and *p* = 0.0335; Figure 1a) [12,21]. However, non-significant differences were observed in *MALINC1* expression levels when DCIS and invasive ductal carcinoma (IDC) samples were compared (*p* = 0.619; Figure 1a) [22]. *MALINC1* expression levels were also compared across DCIS intrinsic subtypes obtained from the GSE69994 dataset. Luminal A and luminal B intrinsic subtypes showed significantly higher levels of *MALINC1* expression compared with the HER2 and basal-like subtypes in pre-invasive samples (*p* < 0.01; Figure 1b).

In order to assess whether *MALINC1* overexpression was associated with cancer progression or patient outcome, *MALINC1* gene expression profiles from primary invasive breast carcinomas obtained from the GDC-TCGA BRCA project (*n* = 1097) were grouped into low or high *MALINC1* expression levels using the StepMiner algorithm [18] (Figure 1c). Interestingly, a significantly larger number of tumors with high *MALINC1* expression were detected in luminal A (82%) and luminal B (80%) subtypes compared with HER2 (44%) and basal-like (25%) breast cancer subtypes (*p* = 1.7 × 10^−24^; Figure 1d). These results are in agreement with the higher *MALINC1* expression observed in luminal DCIS described above (Figure 1b). As shown in Figure 1e, Kaplan–Meier analysis for luminal intrinsic subtype revealed that the subgroup of patients with high *MALINC1* expression was associated with a shorter overall survival (median survival = 10 years) compared with those with low expression (median survival = 18 years) (log-rank *p* = 0.027), whereas non-significant differences were detected in the overall survival of basal-like and HER2+ breast carcinomas (*p* = 0.31) (Appendix A). These results corroborate and extend the previous findings of Bida et al. (2015), who suggested that high levels of *MALINC1* were associated with poor prognosis in 90 breast cancer patients. Taken together, these results suggest that *MALINC1* expression may influence breast cancer progression, remaining up-modulated in primary invasive carcinomas. More importantly, evaluation of *MALINC1* expression levels in primary breast cancer samples could be useful as a biomarker of patient prognosis and outcome.

### 3.2. MALINC1 Is an E2-ER-Modulated Gene Enriched in the Cytoplasmic Fraction of Luminal Breast Cancer Cells

To further evaluate the association and trends of *MALINC1* overexpression in specific breast cancer subytpes, we performed RT-qPCR on different cell lines, as shown in Figure 2a. The highest *MALINC1* expression levels were observed in T47D cells, but also detected in other luminal-like breast cancer cell lines such as ZR75-30 and MCF7. DCIS.COM and normal (MCF10A) cell lines displayed low levels of *MALINC1* expression compared with breast cancer cell lines (*p* = 0.0367). Non-significant difference in *MALINC1* expression was detected between MCF10A and DCIS.COM cells (*p* = 0.3061). Interestingly, the MDA-MB231 basal-like breast cancer cells showed increased expression levels compared with some luminal-like breast cancer cells, such as MCF7 (Figure 2a). Since lncRNAs exert their functions through gene expression modulation and their regulatory manners vary according to the subcellular location, we further explored the *MALINC1* localization in breast cancer cells by RT-qPCR. Cellular homogenates were separated into cytoplasmic and nuclear fractions. LncRNA *MALAT1* and *MTRNR1* (*mitochondrially encoded 12S RNA*) were used as nuclear and cytoplasmic control markers, respectively. T47D and MCF7 displayed a significant enrichment of *MALINC1* in the cytoplasmic fraction (*p* = 0.004 and *p* = 0.029, respectively). However, the DCIS.COM cells showed nuclear and cytoplasmic localization for *MALINC1* lncRNA (Figure 2b). Similar results were reported for U2OS osteosarcoma cells, where *MALINC1* localization was detected in both nuclear and cytoplasmic fractions [15]. Cytoplasmic lncRNAs modulate gene expression and their associated signaling pathway by maintaining cellular structure and functions related to mRNA translation and stability, protein scaffolding, localization, and turnover [23]. In addition, cytoplasmic lncRNAs capable of modulating signal transduction pathways by binding specific signaling molecules and/or altering their phosphorylation status have also been characterized in inflammatory and immune-related processes as in cancer [23]. In this sense, further mechanistic studies of the *MALINC1* interactors may provide specific insights into how this lncRNA modulates gene/protein expression in breast cancer cells.

In the context of breast cancers, estrogens have long been known to promote tumor cell growth through the regulation of several growth-promoting factors such as stimulatory cell cycle-related genes [24,25,26]. Luminal-like breast tumor cells express estrogen receptors alpha and beta (ERα and ERβ, respectively), which are members of the nuclear receptor family of ligand-activated transcription factors that control proliferation, survival, and functional status [27,28]. As *MALINC1* expression was associated with luminal-like breast cancer lines such as T47D and MCF7, which are characterized by their high dependence on estradiol for growth, we reasoned that *MALINC1* overexpression might also be modulated by the E2-ER signaling pathway. To evaluate the expression of *MALINC1* in this hormone-responsive tumor phenotype, a RT-qPCR analysis was performed in 17β-estradiol (E2)-stimulated MCF7 cells at various times post treatment using the *XBP1* gene as an E2-ER responsive reference. Interestingly, we found that E2 treatment increased *MALINC1* expression as early as 1 h after stimulation and reached its peak within 3 h (*p* = 0.006), followed by a marked decrease after 6 h of treatment (Figure 2c). Transcriptional regulation of target genes in response to E2 is mediated by two main mechanisms. In one case, the E2-ER complex binds to a specific DNA sequence called the estrogen response element (ERE), which interacts with co-regulatory proteins, promoting chromatin remodeling and bridging with the general gene transcription machinery, thus resulting in transcription initiation [29]. Alternatively, E2 also exerts rapid, non-genomic effects attributed to cell membrane-initiated signaling [30,31]. To identify the occurrence of EREs within the promoter regions of *MALINC1,* the 1200 bp upstream sequence relative to the TSS was retrieved from UCSC genome browser based on the primer sequences used for *MALINC1* promoter cloning described by Bida et al. (2015) [15] for further sequence analysis using the JASPAR resource (https://jaspar.genereg.net) (accessed on 1 September 2021). As shown in Figure 2d, the in silico analysis predicted four EREs binding sites in the proximity to the *MALINC1* transcriptional start sites (TSS), providing additional support for an E2-ER-mediated modulation of *MALINC1* expression that deserves further characterization.

### 3.3. Transcriptome Analysis of MALINC1-Overexpressing Cells 

To better understand the mechanism of action of *MALINC1* and their phenotypic impact in normal and pre-invasive tumor cells, MCF10A and DCIS.COM cells were stably transduced for *MALINC1* overexpression with a *MALINC1* expression vector (pLOC-*MALINC1*) or with an empty vector (pLOC-empty) for further transcriptomic characterization. RT-qPCR was performed in both transformed cell lines to corroborate *MALINC1* overexpression compared with the empty vector used as control (*p* < 0.001; data not shown). As observed in Figure 3a, the *MALINC1* induction in transduced cells compared to controls was 11.7 (*p* = 0.035) and 14.3 (*p* = 0.005) fold changes in MCF10A and DCIS.COM cells, respectively. Whole-transcriptome unsupervised analysis from RNA-seq data demonstrated a clear segregation of transduced cells in MCF10A and DCIS.COM groups, confirming the different cell types (Figure 3b). RNA-seq analysis of MCF10A cells identified 783 differentially expressed genes (DEGs), of which 411 were upregulated and 372 were downregulated, comparing *MALINC1*-expressing cells with the empty-vector cells (FDR < 0.01 and FC > 2; Figure 3c). On the other hand, in DCIS.COM cells, *MALINC1* overexpression caused the deregulation of 648 genes, of which 214 were upregulated and 434 were downregulated (FDR < 0.01 and FC > 2; Figure 3c). A significant number of genes commonly modulated (62 genes; *p* = 1.63 × 10^−5^, Appendix A) between MCF10A and DCIS.COM were detected among *MALINC1*-overexpressing cells (Figure 3d). This overlap represents a significant 1.7-fold enrichment over the expected number based on random sampling of all expressed genes. Interestingly, enrichment analysis of the transcription factor binding sites (TFBS) present in *MALINC1* commonly modulated genes revealed a striking enrichment for *NFE2*, *AP-1*, *POU1/2F1*, *ATF4*, and *RELA* transcription factors (Figure 3e). 

Functional enrichment analysis of DEGs in MCF10A and DCIS.COM cells showed a significant overrepresentation of genes related to extracellular matrix (ECM) organization, cell adhesion, cell proliferation/division, and several immune-related GO biological processes (*p* < 0.0001) (Figure 3f,g). Genes involved in tumor microenvironment remodeling such as collagens, matrix metalloproteinases (MMP), and cell adhesion molecules are consistently upregulated in DCIS to IDC transition [32]. In addition to the importance of ECM remodeling processes to facilitate cancer invasion, immune microenvironmental changes have been documented as relevant factors promoting early-stage breast cancer progression facilitating immunosurveillance escape of tumor cells [12,33,34]. Furthermore, pathway enrichment analysis of DEGs revealed multiple commonly modulated pathways between normal and DCIS.COM *MALINC1*-transduced cells, such as ECM degradation, integrin signaling pathway, transcriptional targets of AP-1 and deltaNp63, direct TP53 effectors, and TGFB and Wnt signaling pathways, among others (*p* < 0.0001) (Figure 3h). In agreement with these findings, several of these signaling pathways have been previously associated with early-stage breast cancer progression, pointing to TP53 pathway inactivation or aberrant activation of TGFB and Wnt signaling as extremely common events in DCIS samples [12,35,36,37,38]. 

### 3.4. MALINC1 Overexpression Promotes In Vitro Migration of Normal Breast Cells

During DCIS-to-IDC conversion, epithelial cells acquire the ability to infiltrate the surrounding tissues for later dissemination to secondary organs mostly via lymphatic vessels. This process requires the acquisition of novel migratory and invasive properties by the epithelial tumor cells and the ability to remodel and degrade the extracellular matrix environment that acts as a physical barrier to invasion. In this sense, we identified a consistent enrichment of ECM/collagen degradation, epithelial and mesenchymal transition, and cell adhesion bioprocess in *MALINC1*-transduced cells, suggesting the acquisition of the above-described properties. To evaluate the phenotypic impact of *MALINC1* overexpression in cell motility and migration, we conducted wound-healing and transwell migration assays on stably transduced MCF10A cells (Figure 4a,b). As can be observed in Figure 4, stable *MALINC1* overexpression did not seem to enhance cell motility (*p* = 0.483) (Figure 4a), but promoted cell migration in normal breast cells after a week of cell culture (*p* = 3.9 × 10^−5^) (Figure 4b). As the mechanism underlying coordinated regulation of cell motility and migration in epithelial collectives is not yet fully understood, more studies are needed. We finally determined the effects of stable *MALINC1* expression on cell proliferation by means of the MTT assay. Stable *MALINC1* expression did not significantly impact cell proliferation in normal breast epithelial cells after 2, 4, and 6 days of cell culture (*p* > 0.05) (Appendix A). The enhancement of cell migration has been associated with the invasion and metastasis of transformed epithelial cells. In this sense, *MALINC1* overexpression behaved as a pro-tumorigenic stimulus, inducing increased cell migration in normal breast cells.

### 3.5. Conserved MALINC1 Modulated Pathways among Pre-Invasive and Invasive Stages

Pathway-based representation analysis of RNA-seq profiles identified several signaling pathways that differed in their activity among MCF10A (110 IPAs) and DCIS.COM (124 IPAs) *MALINC1*-overexpressing cells (*p*-adj. < 0.01) (Figure 5a; Appendix A). Among the top 10 activated signaling pathways we found *JUN*, *FRA1/JUN*, and *JUN/FOS* ranked first in the activated pathways in both normal MCF10A and DCIS.COM cells (*p* < 10^−4^) (Figure 5b). As JUN, JUNB, and FOS proteins are subunits of the activator protein-1 (AP-1) family of transcription factors, we hypothesized that up-modulation of these transcripts by *MALINC1* might result in AP-1 activation in normal and breast cancer cells. To corroborate this, *MALINC1*, *JUN*, and *FOS* expression levels were evaluated in 33 normal and breast cancer samples by RT-qPCR (Figure 5c,d). First, *MALINC1* expression was detected in 53% of breast carcinomas (10 out of 19 cases) compared with 7% of normal breast tissues (1 out of 14 cases) (*p* = 0.006). Second, 90% of *MALINC1*-expressing cases were ER-positive tumors (9 out 10), whereas 55% of *MALINC1* non-expressing cases were ER-negative tumors (5 out 9) (*p* = 0.033) (Figure 5d). These results validate the *MALINC1* in silico analysis in normal and breast cancer cells and their association with luminal-like breast cancer subtypes. Second, significant positive correlations were detected between *MALINC1* vs. *JUN* (*r* = 0.67; *p* = 0.001) and *MALINC1* vs. *FOS* (*r* = 0.41; *p* = 0.017) expression levels. More importantly, 100% of high *MALINC1* expression samples were associated with high *JUN* expression compared with 19% of samples with undetectable *MALINC1* expression levels (*p* < 0.0001) (Figure 5d). These results corroborate our findings and suggest that *MALINC1* may modulate JUN and FOS gene expression and the AP-1 complex activity in early stages of breast cancer and invasive breast carcinomas.

AP-1 transcription factors are involved in the transcriptional modulation of several genes associated with differentiation, apoptosis, oncogenic transformation, proliferation, and cell migration [39,40,41]. Aberrant expression of *JUN*, *FOS*, *FRA1*, and *ATF2* has been implicated in breast cancer development and progression [42,43,44]. Expression of AP-1 complex proteins is induced by different environmental signals, such as growth factors, cytokines, UV irradiation, and pathogens [45]. Importantly, the AP-1 complex has specific roles in the immune system, such as T-cell activation, Th differentiation, T-cell anergy, and exhaustion [46,47,48]. For example, Qiao et al. (2016) demonstrated that c-Jun is an important regulator of TNFα-driven transcriptional events, and that increased *JUN* expression is responsible for inflammation-induced malignant processes and the establishment and progression of basal-like breast cancer [49]. Additionally, TNFα enhances luminal breast cancer cell proliferation by inducing aromatase gene expression, allowing high levels of estradiol through c-Fos and c-Jun [50,51]. Interestingly, AP-1 activity in combination with NFAT1, a key regulator of T-cell activation, drives expression of many cytokines involved in immune effector responses [52,53]. More importantly, several AP-1 family members such as JUN, JUNB, and FOS were found to transcriptionally induce the expression of co-inhibitory immune checkpoint genes such as PD-1 and PD-L1 [54,55]. 

Therefore, the evidence presented here suggests that an E2/ER-*MALINC1*-AP1 axis may participate in the modulation of the identified signaling pathways promoting breast cancer progression at pre-invasive stages through tumor and immune microenvironment changes.

### 3.6. Immune Features Associated with MALINC1 Overexpression in Invasive Carcinomas

Several studies suggest that concurrent tumor-specific and immune microenvironmental changes play major roles in the DCIS-to-IDC transition [12,34,56]. Gil Del Alcazar et al. (2017) demonstrated that basal-like and HER2+ DCIS displayed an activated immune environment compared with their invasive counterparts, which are characterized by an immunosuppressive microenvironment with higher PD-L1 expression and Treg cells [56]. Interestingly, functional analysis of DEGs in MCF10A and DCIS.COM *MALINC1*-overexpressing cells revealed the enrichment of genes strongly related to innate immune response (*p* = 3.4 × 10^−7^), immune response (*p* = 3.1 × 10^−6^), inflammatory response (*p* = 2.1 × 10^−5^), and leukocyte cell–cell adhesion (*p* = 1.5 × 10^−3^) (Figure 3f,g). To further explore the impact of *MALINC1* over tumor-infiltrating immune cells, we estimated the immune-cell fractions of high- and low-*MALINC1*-expressing tumors derived from the TCGA-BRCA dataset using the quanTIseq deconvolution algorithm based on their RNA-seq profiles. Increased *MALINC1* expression in luminal-like carcinomas was characterized by a protumorigenic Th2/humoral immunity, as evidenced by the highest fractions of B cells, Treg cells, macrophage M2, and high *PD-L1* expression levels compared with low-*MALINC1*-expression tumors (*p* < 0.05) (Figure 6a). Suppression of antitumor immune response by inducing T-cell anergy due to Th2-polarized activity and/or expansion of Treg cells and increased PD-L1 expression with a subsequent loss of T-cell-mediated cytotoxicity, together with tissue remodeling and the enhancement of cell migration associated with *MALINC1* overexpression, could be instrumental to promoting the progression of DCIS to the infiltrating stages.

Immunotherapy represented by immune checkpoint inhibitors (ICIs) has been largely explored in breast cancer patients, including both early and advanced disease [57]. Luminal-like breast cancer is characterized as a poorly immunogenic subtype with lower PD-L1 expression and tumor-infiltrating immune cells compared with basal-like breast carcinomas. Despite studies that have shown divergent results, PD-L1 has been correlated with worse clinicopathological parameters and poor outcomes in patients with luminal-like breast carcinomas [58]. In recent studies evaluating patients with early-stage luminal-like breast carcinomas, PD-L1 expression was reported at around 9% in luminal A subtype and was increased to about 42% in luminal B [59]. Although immunotherapy and immune combination therapy have also been increasingly explored in luminal-like breast cancer, only a fraction of patients could be sensitive to these therapeutic approaches. Therefore, we explored the role of *MALINC1* expression as a predictive biomarker of ICI response in luminal-like breast carcinomas. To this end, the Tumor Immune Dysfunction and Exclusion (TIDE score) was computed and compared across high- and low-*MALINC1*-expression tumors as a surrogate signature to predict ICI response [20]. High-*MALINC1*-expression tumors were significantly enriched in ICI responder patients (30.5%), as predicted by TIDE score, compared with low-*MALINC1*-expression cases (18%; *p* = 0.0257) (Figure 6b). In addition, high-*MALINC1* ICI responder cases were significantly enriched by luminal B carcinomas (44%) compared with their non-responder counterparts (28%; *p* = 0.0327). Although high *MALINC1* expression was detected in only 25% of basal-like carcinomas, 41% of these tumors were predicted as ICI responder patients compared with 15% of ICI responder cases in the low-*MALINC1*-expression group (*p* = 0.0122) (Figure 6c).

## 4. Conclusions

In conclusion, the described results indicate that *MALINC1* overexpression induced premalignant changes mainly associated with tumor microenvironment remodeling processes and the activation of several protumorigenic signaling pathways such as AP-1 and cell migration in normal, pre-invasive, and invasive breast carcinomas. In addition, we determined that *MALINC1* behaves as an E2-ER-modulated transcript predominantly found in the cytoplasmic compartment of luminal-like breast cells that may influence breast cancer progression, affecting patient outcome. Furthermore, the immune profiling described in our study suggests that high-*MALINC1*-overexpressing cells at pre-invasive and invasive stages are characterized by a tumor-associated immunosuppressive phenotype in luminal-like breast carcinomas. More importantly, *MALINC1* expression behaves as a predictive biomarker of immunotherapy response in luminal and basal-like primary invasive carcinomas that deserves further characterization. Overall, our findings indicate that *MALINC1* is a novel oncogenic and immune-related lncRNA involved in early-stage breast cancer progression.

## Figures and Tables

**Figure 1 cancers-14-02819-f001:**
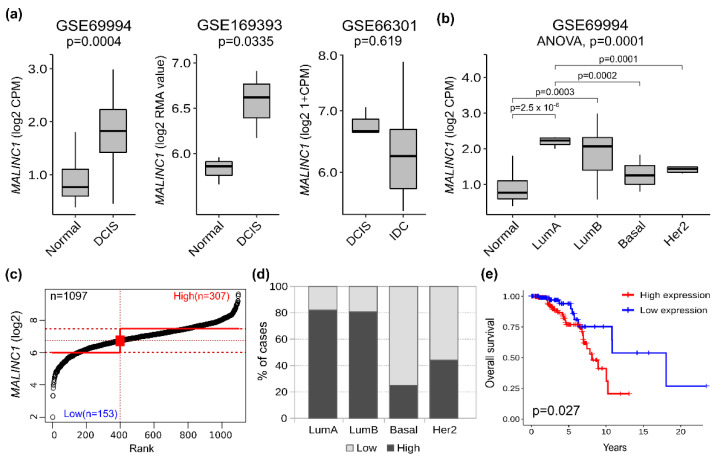
*MALINC1* expression in normal, pre-invasive, and invasive breast samples. (**a**) In silico analysis of *MALINC1* expression among normal, DCIS, and IDC samples obtained from three independent GEO datasets [12,21,22]. *MALINC1* expression (represented in log2 count per million (CPM) or log2 robust multi-array average values (RMA)) was significantly upregulated in DCIS samples compared with normal samples (*p* = 0.0004 and *p* = 0.0335), whereas non-significant differences were observed between DCIS and IDC cases (*p* = 0.619). (**b**) *MALINC1* expression analysis of intrinsic subtypes among normal and DCIS samples obtained from GSE69994 dataset [12]. Data are shown as means ±S.D. (**c**) Primary breast carcinomas were divided into *MALINC1* low or high expression levels based on the StepMiner algorithm using TCGA RNA-seq datasets obtained from the UCSC Xena resource (https://xenabrowser.net/) (accessed on 12 October 2021). (**d**) Percentage of cases with high or low *MALINC1* expression among intrinsic subtypes showing a consistent upregulation in luminal-like tumors compared with basal-like and HER2 subtypes. (**e**) Kaplan-Meier survival analysis on data of 298 patients with luminal-like tumors obtained from the TCGA RNA-seq datasets. Breast cancer patients with high *MALINC1* expression (red line) showed reduced overall survival compared to patients with low expression (blue line) (log-rank *p* = 0.027).

**Figure 2 cancers-14-02819-f002:**
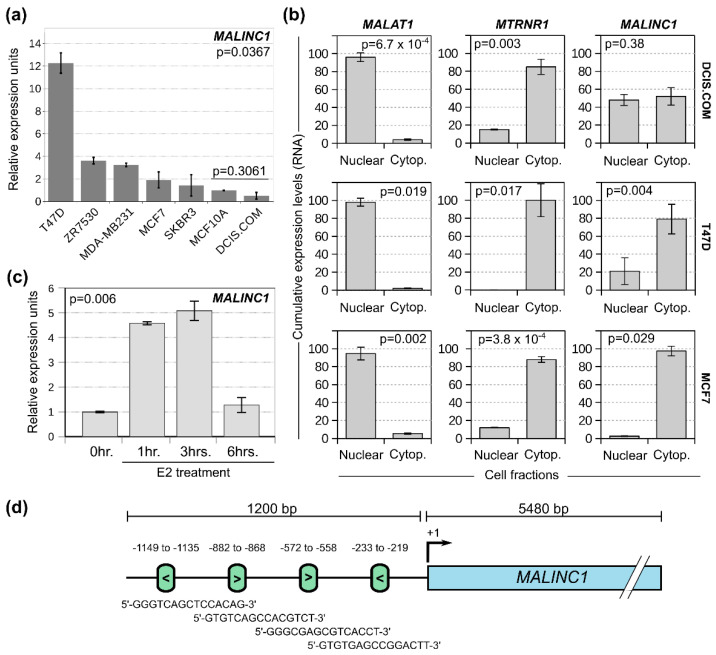
Characterization of *MALINC1* expression and subcellular localization in breast cancer cell lines. (**a**) *MALINC1* expression levels in normal, DCIS, and breast cancer cell lines as determined by RT-qPCR. All assays were performed in triplicate and normalized to housekeeping gene *GAPDH*. Gene expression was expressed as relative units compared to MCF10A normal cell line. (**b**) Subcellular localization of *MALINC1* in DCIS.COM, T47D and MCF7 cells. *MALAT1* and *MTRNR1* were analyzed as nuclear and cytoplasmic markers, respectively. Cellular homogenates were separated into nuclear and cytoplasmic fractions and relative *MALINC1* expression was evaluated by RT-qPCR. T-test was used to compare *MALINC1* expression among cell fractions. (**c**) *MALINC1* expression induction in estrogen-stimulated MCF7 cells at various times post E2 treatment (1, 3, and 6 h). ANOVA with post-hoc Tukey HSD test was employed to compare the different time points. (**d**) Schematic representation of *MALINC1* promoter region with the ESR2 transcription factor binding site (green boxes) predicted by the JASPAR resource (https://jaspar.genereg.net) (accessed on 1 September 2021). Arrows inside green boxes indicate the DNA strand of the mapped transcription factor binding sites.

**Figure 3 cancers-14-02819-f003:**
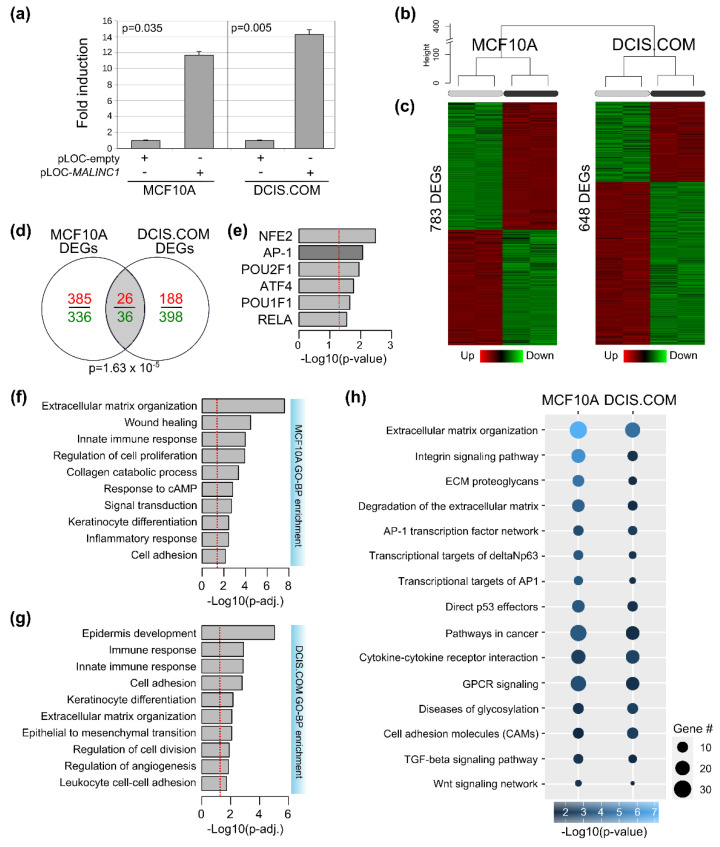
Transcriptomic analysis of *MALINC1*-overexpressing cells. (**a**) Fold induction levels of *MALINC1* expression in stably transduced MCF10A and DCIS.COM cell lines as determined by RNA-seq profiles (log2CPM) in pLOC-*MALINC1* or pLOC-empty transfected cells (*p* = 0.035 and *p* = 0.005, respectively). (**b**) Hierarchical clustering of MCF10A and DCIS.COM stably transduced cells with either vector control (pLOC-empty) or lentivirus expressing *MALINC1* (pLOC-*MALINC1*) based on RNA-seq profiles. (**c**) Heat map representation of the differentially expressed genes (DEGs) obtained by RNA-seq analysis (FDR < 0.01; FC > 2). Red and green colors represent upregulated and downregulated genes, respectively. (**d**) Venn diagram of transcripts commonly modulated among MCF10 and DCIS.COM stably transduced cells. (**e**) Functional enrichment analysis of the transcription factor binding sites (TFBS) in the promoter region of the commonly modulated genes among normal and DCIS cells. Statistical significance was determined by the hypergeometric test. (**f**,**g**) Functional enrichment of bioprocesses identified as affected by the expression of *MALINC1* in MCF10A and DCIS.COM cells. At the top, bioprocesses are enriched due *MALINC1* overexpression in MCF10A cell lines. At the bottom, bioprocesses are enriched due *MALINC1* overexpression in DCIS.COM cell lines. The red dotted line represents the basic significance level (*p* < 0.05). (**h**) Pathway commonly affected among MCF10A and DCIS.COM *MALINC1* stably transduced cells (*p* < 0.05).

**Figure 4 cancers-14-02819-f004:**
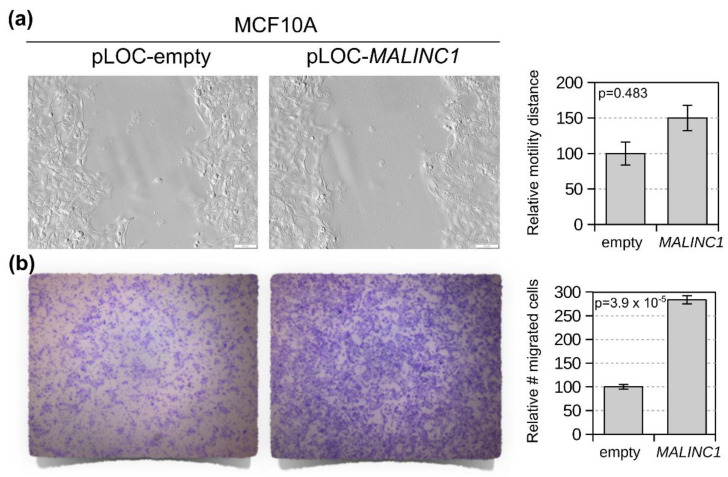
Stable *MALINC1* overexpression effect in cell motility and migration of normal breast cells. (**a**) MCF10A-transduced cells with either vector control or lentivirus expressing *MALINC1* were compared using the in vitro wound-healing assay. Forty-eight hours after the original scratch the area covered by migrating cells from the edges was compared. As can be observed in representative images, non-significant difference in cell motility was observed (*p* = 0.483). Scale bar = 400 μm. (**b**) Transwell migration assay of MCF10A cells stably transduced with *MALINC1*. On the left, comparative pictures of cells that migrated through the membrane; on the right, a box-and-whisker plot of numbers of cells per membrane (*p* = 3.9 × 10^−5^). Statistical significance was determined using the *t*-test.

**Figure 5 cancers-14-02819-f005:**
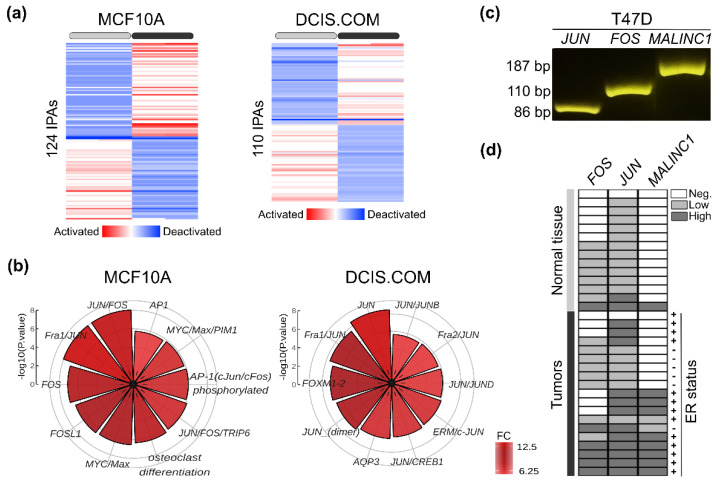
Pathway activity analysis in normal and DCIS *MALINC1*-transduced cells. (**a**) Heatmap of integrated pathways activities (IPAs) differentially modulated in *MALINC1*-overexpressing cells as predicted by PARADIGM using the normalized RNA-seq expression profiles (p-adj. < 0.01). (**b**) Circular bar plot of the top 10 most significantly activated IPAs in MCF10A and DCIS.COM *MALINC1*-overexpressing cell lines. (**c**,**d**) Agarose gel of *MALINC1*, *FOS*, and *JUN* RT-PCR amplicons in the T47D luminal-like breast cancer cell line. (**d**) Evaluation of *MALINC1*, *FOS*, and *JUN* expression levels by RT-qPCR on breast normal and tumor samples. Gene expression levels were normalized to the RNA18S transcript and discretized in undetectable (negative in white boxes), low-, or high-*MALINC1*-expressing samples (gray or black boxes, respectively).

**Figure 6 cancers-14-02819-f006:**
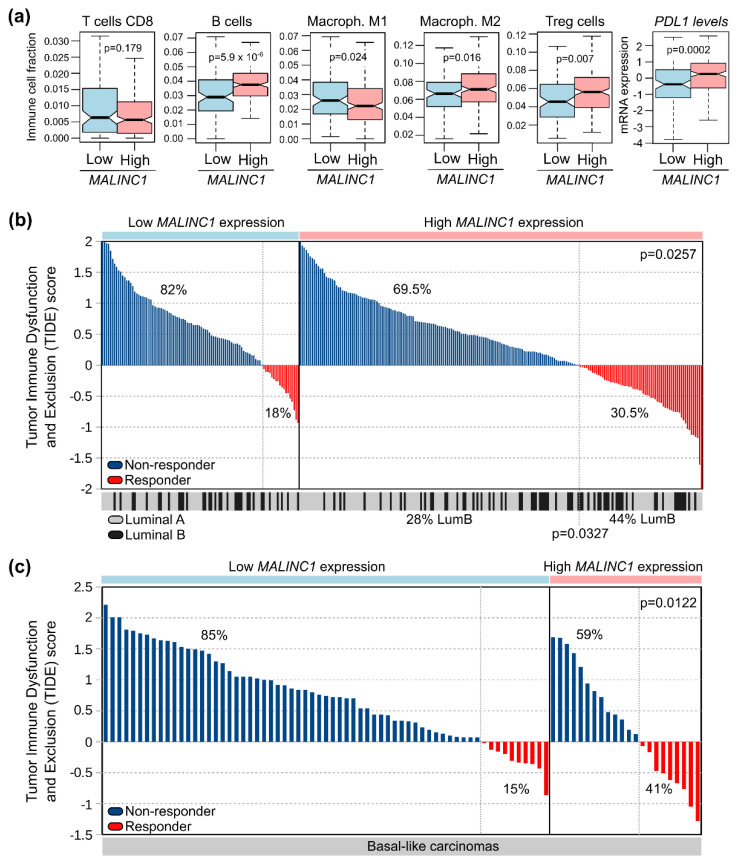
Comparative immune profiling of high- and low-*MALINC1*-expressing breast carcinomas. (**a**) Immune-cell fractions and *PDL1* expression of high and low *MALINC1* luminal-like tumors obtained from the TCGA-BRCA project as estimated by the quanTIseq deconvolution algorithm based on RNA-seq profiles. Statistical differences were determined using a Mann-Whitney-Wilcoxon test. (**b**,**c**) Waterfall plots of TIDE prediction scores across tumors with high or low *MALINC1* expression in luminal-like and basal-like breast carcinomas. Breast cancer samples were sorted from high to low TIDE scores for their classification in non-responder (positive values in blue) and responder cases (negative values in red), as suggested by the TIDE resource.

## Data Availability

Transcriptomic data have been deposited in the Gene Expression Omnibus database (accession number GSE194150).

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
