# Peer review of "MALINC1 an Immune-Related Long Non-Coding RNA Associated with Early-Stage Breast Cancer Progression"

_cancers, 2022, doi:10.3390/cancers14122819_

Round 1
Reviewer 1 Report
The authors are investigating the role of the lncRNA MALINC1 in invasive vs non-invasive breast cancer models. They identify it as being over-expressed in both pre-invasive and invasive breast cancer samples, where it is associated with poorer prognosis. They identified MALINC1 as a predominantly cytoplasmic, ER-regulated transcript. Though over-expression RNAseq analysis they identified a MALINC1-dependent transcriptional profile enriched for ECM modulators and cell adhesion, and identified the JUN/FOS pathway as the most highly activated after MALINC1 over expression.
Major Concerns:
1) The MALINC1 dependent gene profiling, pathway analysis and subsequent in vitro analyses of motility and invasiveness are all conducted in cells over-expressing MALINC1. The fold induction of MALINC1 after infection and selection is reported as being 450x (MCF10A) and 45000x (DCIS.COM) in Figure 3a. Given that the most highly expressing breast cell tested, T47D, has approximately 12-fold higher expression than MCF10A (Figure 2a), and that maximal induction of MALINC1 by E2 in MCF7 cells is ~5x (Figure 2c), these over expression experiments are many orders of magnitude above the range of physiological expression. The level of lncRNA needs to be titrated down to reflect what might be considered physiological expression levels. Data derived from MALINC1 knockdown in T47D and/or MDA-MB231 cells, where transcript levels can be reasonably manipulated within the expected physiological range, must also be included to confirm findings from exogenous over-expression.
2) MALINC1 has been previously reported to regulate cell cycle progression (PMID: 26337085) where the correlation between high MALINC1 expression and poor prognosis in breast cancer was reported. The effect of MALINC1 on cell cycling in the cell lines used in this study should be explored.
3) absolute p-values should be given in all instances.
Author Response
We thank the reviewer for their effort in reviewing our manuscript, and their constructive criticisms and suggestions.
Response to Reviewer #1
1. Reviewer: The MALINC1 dependent gene profiling, pathway analysis and subsequent in vitro analyses of motility and invasiveness are all conducted in cells over-expressing MALINC1. The fold induction of MALINC1 after infection and selection is reported as being 450x (MCF10A) and 45000x (DCIS.COM) in Figure 3a. Given that the most highly expressing breast cell tested, T47D, has approximately 12-fold higher expression than MCF10A (Figure 2a), and that maximal induction of MALINC1 by E2 in MCF7 cells is ~5x (Figure 2c), these over expression experiments are many orders of magnitude above the range of physiological expression. The level of lncRNA needs to be titrated down to reflect what might be considered physiological expression levels. Data derived from MALINC1 knockdown in T47D and/or MDA-MB231 cells, where transcript levels can be reasonably manipulated within the expected physiological range, must also be included to confirm findings from exogenous over-expression.
Author response: We thank the reviewer for his/her detailed observations, and we agree that the semiquantitative qRT-PCR analysis previously shown in Figure 3a only corroborates the MALINC1 overexpression in MCF10A and DCIS.COM stably transduced cells. Therefore, we have replaced the previous qRT-PCR based Figure 3a by the MALINC1 fold induction based on the quantitative counts per million values obtained from the RNA-seq profile to better define the MALINC1 induction levels reached by the transduced cells. As observed in the current Figure 3a, the MALINC1 induction in transduced cells is approximately 12-14 fold changes compared to controls, reflecting physiological expression levels in the context of previously performed assays (Figures 2a and 2c). Regarding the suggested MALINC1 knockdown assay in DCIS or invasive breast cancer cells, we must state that is a non-trivial approach due to lncRNAs localized in both nuclear and cytoplasmic compartments such as MALINC1 requires a mixed-modality approach combining ASOs and RNAi reagents for an effective suppression (cytoplasmic lncRNAs are suppressed using RNAi and nuclear ones are more effectively suppressed using ASOs [PMID:26578588]). It is worth noting however that our report is primarily focused on the role of MALINC1 in early stages of breast cancer progression and not in invasive breast cancer lines in which many other genomic/transcriptomic abnormalities have taken place. Importantly, whatever findings could be obtained by silencing MALINC1 in breast cancer lines could be non-comparable to effects in MCF10A and even non-comparable between cancer lines , i.e. the effects of silencing MALINC1 in MDAMB231 not necessarily will compare (phenotypically, transcriptomically) with whichever effects could occur in T47D cells. We strongly believe that the conclusions of our study are consistently supported by the experimental evidence provided.
2. Reviewer: MALINC1 has been previously reported to regulate cell cycle progression (PMID: 26337085) where the correlation between high MALINC1 expression and poor prognosis in breast cancer was reported. The effect of MALINC1 on cell cycling in the cell lines used in this study should be explored.
Author response: As suggested by the reviewer, we introduced a new assay to compare the proliferative properties of MCF10A MALINC1 stably transduced cells. Although, Bida et.al, 2015 showed that reducing the endogenous MALINC1 RNA levels in invasive U2OS osteosarcoma cells resulted in cell cycle redistribution specifically inhibits M phase exit and cell cycle progression, our results suggest that MALINC1 overexpression does not significantly increase cell proliferation in normal breast epithelial cells (see Materials & Methods section on page 3 and Results & Discussion section on page 10).
3. Reviewer: Absolute p-values should be given in all instances.
Author response: As suggested by the reviewer, absolute p-values were introduced in all instances.
Reviewer 2 Report
This study by Abba and team is well designed and presented clearly. Below are some of my comments, which may improve data quality and overall analysis.
Fig 1.
- Line 215-217 “While non-215 significant differences were detected in the overall survival of basal-like and HER2+ breast 216 carcinomas (p>0.05).”—Figure or data not mentioned.
- Fig 1a, what is the unit of MALINC1 expression?
- What about the comparison of normal vs. IDC?
- Although DCIS vs. IDC does not show any significant difference in MALINC1 expression, a lower expression trend appears in IDC based on Q2 and Q3 expression values.
- Fig 1b, Does the GSE169393 dataset also show a similar profile of MALINC1 expression? What is the unit of MALINC1 expression?
Fig 2.
- Line 257, “MALINC1 localization is either nuclear or cytoplasmic.” Does the author mean “both” and not “either”?
- Fig 2a showed higher expression of MALINC1 in normal cells, i.e., MCF10A compared to DCIS. Please explain? This graph also showed that not all Luminal cell line has higher MALINC1 expression. This fact needs to be mentioned in the text.
- Fig 2b, How come the cumulative expression level is more than 100% in Nuc + cytoplasmic level? The combined values of Nuc+ Cyto should be 100%. Please correct.
- Fig 2d, How did the author confirmed the promoter region of MALINC1? What are the arrows in the binding regions indicate?
- Authors have shown that E2 modulates the expression of MALINC1 in MCF7 cells, which were positive for ER. Furthermore, they identified an in-silico E2-ER binding site upstream of the MALINC1 gene, arguing a potential regulatory mechanism but have not characterized it further. In this case, it becomes crucial to perform a similar experiment with E2 (as in fig 2c) in ER-negative cell lines to validate this hypothesis.
Fig 3.
- In my opinion, analysis of genes and pathways specific to MCF10A and DCIS.COM upon MALINC1-OE would be more informative to understand the biology rather common genes since these two lines represent normal and cancerous properties, and that could be differently regulating signaling pathways. This is just an observational comment.
Fig 4.
- Authors have mentioned that they tested the invasion ability of MCF10A upon MALINC1 OE in lines 369-370. But they have not provided invasion data. Please provide.
- Authors should knockdown MALINC1 in T47D and show the opposite to confirm the role of MALINC1.
- What happens when MALINC1 gets OE in DCIS.COM?
Fig 5.
- Does the level of JUN and FOS change upon OE and KD of MALINC1?
Minor:
- Line 146 - analisis should be analysis.
- Line 175 – simples should be samples
Author Response
We thank the reviewer for their effort in reviewing our manuscript and their constructive criticisms, and considering it as a well designed and clearly presented study.
Response to Reviewer #2
- Reviewer: Fig 1. Line 215-217 “While non-215 significant differences were detected in the overall survival of basal-like and HER2+ breast 216 carcinomas (p>0.05).”—Figure or data not mentioned.
Author response: As suggested by the reviewer, a new Supplementary Figure S1 was introduced with the Kaplan–Meier curves of basal-like and HER2+ breast cancer patients.
- Reviewer: Fig 1a, what is the unit of MALINC1 expression?
Author response: As required by the reviewer, the MALINC1 expression units were included in each boxplot of the Fig1a. and described in the Fig1a legend (See highlighted paragraph on page 5). We apologize for this oversight.
- Reviewer: Fig 1a, What about the comparison of normal vs. IDC?
Author response: In response to this concern, we must state that the data mining approach was focused on the normal-DCIS and DCIS-IDC transitions of breast cancer progression. In addition, a very limited number of breast cancer profiling series (the included ones) have been properly mapped and annotated for MALINC1 lncRNA (ENSG00000245146).
- Reviewer: Fig1a, Although DCIS vs. IDC does not show any significant difference in MALINC1 expression, a lower expression trend appears in IDC based on Q2 and Q3 expression values.
Author response: As observed by the reviewer in the GSE66301 dataset, MALINC1 expression showed a non-significant difference between DCIS and IDC groups (p=0.619). The highest variability in MALINC1 expression among IDC samples, and the trend to a lower MALINC1 expression compared to DCIS samples, could be explained by the tumor intrinsic subtypes composition. Unfortunately, the tumor intrinsic subtype classification was not provided within the GSE66301 dataset for further evaluation.
- Reviewer: Fig 1b, Does the GSE169393 dataset also show a similar profile of MALINC1 expression? What is the unit of MALINC1 expression?
Author response: The MALINC1 expression units are now included in Fig 1b. The GSE169393 Affymetrix-based dataset represents information on samples from three DCIS patients from whom DCIS specimens and non-tumorous normal regions of the mammary gland were obtained. Additionally, the intrinsic breast cancer subtypes or hormone receptor status of these samples was not described for further evaluation.
- Reviewer: Fig 2, Line 257, “MALINC1 localization is either nuclear or cytoplasmic.” Does the author mean “both” and not “either”?
Author response: The refereed paragraph was modified as follows: "Similar results were reported for U2OS osteosarcoma cells where MALINC1 localization was detected in both nuclear and cytoplasmic fractions [15]."
- Reviewer: Fig 2a showed higher expression of MALINC1 in normal cells, i.e., MCF10A compared to DCIS. Please explain? This graph also showed that not all Luminal cell line has higher MALINC1 expression. This fact needs to be mentioned in the text.
Author response: As shown in the current version of Figure 2b, where comparative p-values among cell lines are now included, MCF10A and DCIS.COM cells showed non-significant difference in MALINC1 expression as determined by qRT-PCR (p>0.05). In agreement, non-significant differences were detected in MALINC1 expression levels when MCF10A (0.25±0.19 log2CPM) and DCIS.COM (0.38±0.16 log2CPM) control cells (cells transduced with empty vector) were compared based on RNA-seq data (p>0.05) (Figure 3a). Also as the reviewer suggested we introduced a paragraph in the Results and Discussion section (see highlighted paragraph on page 6) to mention the above described results and the fact that not all luminal cell lines have increased MALINC1 expression compared to other subtypes just as the MDA-MB231 cancer cells.
- Reviewer: Fig 2b, How come the cumulative expression level is more than 100% in Nuc + cytoplasmic level? The combined values of Nuc+ Cyto should be 100%. Please correct.
Author response: Figure 2b has been corrected to represent the cell fraction markers and MALINC1 lncRNA in terms of cumulative expression levels. We apologize for this oversight.
- Reviewer: Fig 2d, How did the author confirmed the promoter region of MALINC1? What are the arrows in the binding regions indicate?
Author response: The promoter region of MALINC1 (1200 bp upstream the TSS) was retrieved from UCSC genome browser according to the primers sequences described in Bida et al., 2015. A brief paragraph was introduced in the Results and Discussion section (see highlighted paragraph at the end of page 7) to describe how the MALINC1 promoter region was obtained. In addition, Figure 2d legend was modified to mention that arrows in green boxes indicate the DNA strand of the estrogen response element identified (see highlighted paragraph on page 7).
- Reviewer: Authors have shown that E2 modulates the expression of MALINC1 in MCF7 cells, which were positive for ER. Furthermore, they identified an in-silico E2-ER binding site upstream of the MALINC1 gene, arguing a potential regulatory mechanism but have not characterized it further. In this case, it becomes crucial to perform a similar experiment with E2 (as in Figure 2c) in ER-negative cell lines to validate this hypothesis.
Author response: Gene expression response to E2-ER signaling is usually assessed in ER responsive cells with their respective controls (including 17β-estradiol untreated cells as negative/unresponsive control). However, it is not rational to include in this assay ER-negative breast cancer cells that by definition are unresponsive to the E2-ER signaling. Nevertheless, further mechanistic studies will be conducted to better define the ER-mediated transcriptional regulation of MALINC1 expression in luminal-like breast cancer cells.
- Reviewer: Fig 3, In my opinion, analysis of genes and pathways specific to MCF10A and DCIS.COM upon MALINC1-OE would be more informative to understand the biology rather common genes since these two lines represent normal and cancerous properties, and that could be differently regulating signaling pathways. This is just an observational comment.
Author response: We thank the reviewer's comment. The MALINC1 transcriptomic data mining approach was designed on the commonly modulated genes/pathways among MCF10A and DCIS.COM stably transduced cells due they have a closely related genomics background. Indeed, the DCIS.COM cell line was derived from a xenograft originating from MCF10A cells (https://doi.org/10.1093/jnci/92.14.1185a). This strategy has the potential to narrow the finding identified in each cell line separately to facilitate the identification of the most consistent bioprocesses modulated by MALINC1 lncRNA. However, the readers can find all transcripts differentially expressed in MCF10A and DCIS.COM cell lines in Supplementary data 1.
- Reviewer: Fig 4, Authors have mentioned that they tested the invasion ability of MCF10A upon MALINC1 OE in lines 369-370. But they have not provided invasion data. Please provide. What happens when MALINC1 gets OE in DCIS.COM? Authors should knockdown MALINC1 in T47D and show the opposite to confirm the role of MALINC1.
Author response: The phenotypic effects (cell proliferation, motility, and migration) were tested in MCF10A with the aim to evaluate the lncRNA impact in normal breast epithelial cells. We apologize for the mistake; the term “cell invasion” was replaced by “cell migration” in the referred paragraph. Regarding MALINC1 knockdown assay in T47D cells, we must state that abrogation of lncRNA localized in both nuclear and cytoplasmic compartments is not trivial and requires a mixed-modality approach combining ASOs and RNAi sequences for an effective suppression [PMID:26578588]. More importantly, whatever findings could be obtained by silencing MALINC1 in invasive breast cancer lines could be non-comparable to effects in normal cells, i.e. the effects of silencing MALINC1 in MCF10A not necessarily will compare (phenotypically, transcriptomically) with whichever effects could occur in T47D cells. We strongly believe that the conclusions of our study are consistently supported by the experimental evidence provided.
- Reviewer: Fig 5, Does the level of JUN and FOS change upon OE and KD of MALINC1?
Author response: Indeed, several genes that encode AP-1 transcription factor subunits such as FOS, FOSB, FOS like 1 and 2 were up-modulated in RNA-seq data of MCF10A and DCIS.COM MALINC1 overexpressing cells as reported in Supplementary data 1. In addition, we must state that PARADIGM pathway analysis produces a data matrix of integrated pathway activities (IPAs) derived from NCI-PID, BioCarta and Reactome pathway databases, based on primary and secondary effectors in the network structures.
- Reviewer: Minor, Line 146 - analisis should be analysis.
Line 175 – simples should be samples
Author response: Lines 146 and 175 were accordingly modified.
Round 2
Reviewer 1 Report
1) qRT-PCR is by definition, quantitative. The analysis of RNAseq data in the revised figure 3a reveals a large discrepancy between the initially reported fold induction after infection and selection as analyzed by qRT-PCR (original figure 3a), and this discrepancy must be addressed.
To determine relevant physiological levels of MALINC1, the lncRNA copy number per cell for each cell line and stable cell line should be determined by droplet-digital PCR or by qRT-PCR comparison to in vitro transcribed RNA. This would allow for direct comparison of the expression level of MALINC1 between these experiments.
MALINC1 has already been reported to be successfully knocked down by siRNA in U2OS cells (https://www.oncotarget.com/article/4944/text/). There are at least 2 other technologies that could be employed to knockdown lncRNAs in cell lines. CasRx (RfxCas13d) has been shown to efficiently target lncRNA in both the nucleus and cytoplasm, and CRISPRi can be used to suppress expression. (PMID: 33685382).
2) While a sentence has been included in the text to address the effect of MALINC1 over-expression on proliferation (line 390-393), data is not shown and cannot be assessed.
3) Absolute p values have not been updated in the entire manuscript. In most instances in sections 3.1-3.4, Fig 1, Fig 2, Fig 3 and Fig 4 p-values are given as p<0.05 or p<0.01.
Author Response
Response to Reviewer #1
1. Reviewer: qRT-PCR is by definition, quantitative. The analysis of RNAseq data in the revised figure 3a reveals a large discrepancy between the initially reported fold induction after infection and selection as analyzed by qRT-PCR (original figure 3a), and this discrepancy must be addressed.
To determine relevant physiological levels of MALINC1, the lncRNA copy number per cell for each cell line and stable cell line should be determined by droplet-digital PCR or by qRT-PCR comparison to in vitro transcribed RNA. This would allow for direct comparison of the expression level of MALINC1 between these experiments.
Author response: The semiquantitative RT-qPCR analysis was used to validate MALINC1 overexpression in MCF10A and DCIS.COM stable transduced cells. The discrepancy -in terms of fold inductions- with the quantitative RNA-seq data are probably due to the RT-qPCR expression levels were obtained in ‘relative expression units’, while the RNA-seq profiling identify the absolute number of transcript counts per sample avoiding the use of reference gene for normalization. Therefore, the RNA-seq based data was employed to better define the MALINC1 induction levels reached by the transduced cells, as shown in Figure 3a. Although the use of droplet-digital PCR approach is a valid suggestion, we think that further characterization of MALINC1 fold induction using this method does not really add much to the main conclusions of the manuscript.
2. Reviewer: MALINC1 has already been reported to be successfully knocked down by siRNA in U2OS cells (https://www.oncotarget.com/article/4944/text/). There are at least 2 other technologies that could be employed to knockdown lncRNAs in cell lines. CasRx (RfxCas13d) has been shown to efficiently target lncRNA in both the nucleus and cytoplasm, and CRISPRi can be used to suppress expression. (PMID: 33685382).
Author response: We thank the reviewer’s advice regarding knockdown assays in invasive breast cancer cells. As we previously stated, our report was primarily focused on the role of MALINC1 in normal and DCIS cells and not in invasive breast cancer lines in which many other mutational and transcriptional changes have taken place. The suggested assays will be implemented in further studies to better define the mechanistic role of MALINC1 in advanced stages of breast cancer progression.
3. Reviewer: While a sentence has been included in the text to address the effect of MALINC1 over-expression on proliferation (lines 390-393), data is not shown and cannot be assessed.
Author response: The result of the cell proliferation assay on MCF10A MALINC1 overexpressing cells was introduced in Supplementary Figure S2.
4. Reviewer: Absolute p values have not been updated in the entire manuscript. In most instances in sections 3.1-3.4, Fig 1, Fig 2, Fig 3 and Fig 4 p-values are given as p<0.05 or p<0.01.
Author response: We apologize for misunderstood the reviewer suggestion. Absolute p-values were now introduced in all instances of the manuscript.
Reviewer 2 Report
Authors have responded to my comments except
Comment 12: what happens upon MALINC1 OE in DCIS.COM?
Also, I am not sure what they meant by "invasion" replaced by "migration" because I don't see any data regarding invasion assay. If you are not showing any data corresponding to invasion, then please remove it from the text section. Or show the data.
There is some issue at Y-axis in Fig 2a.
Author Response
Response to Reviewer #2
1. Reviewer. Comment 12: what happens upon MALINC1 OE in DCIS.COM?
Also, I am not sure what they meant by "invasion" replaced by "migration" because I don't see any data regarding invasion assay. If you are not showing any data corresponding to invasion, then please remove it from the text section. Or show the data.
Author response: As was previously stated, the MALINC1 phenotypic effects were tested in MCF10A normal breast cancer cells, but were not assessed in DCIS.COM cells. We performed a cell migration assay and not an invasion assay, and for that, the term ´invasion´ was properly removed from the text.
2. Reviewer. There is some issue at Y-axis in Fig 2a.
Author response: The reported issue in Fig 2a. was due to the conversion process from .doc to .pdf file provided by the journal website. The originally uploaded figures are correct.